# Uncovering Treatment-Responsive Subgroups Under Direct Bias Control

## Abstract

Identifying subpopulations that benefit most or least from a treatment is central to scientific research and policy analysis. We propose an optimization-based framework for learning such subgroups from observational data. The proposed methods discover subgroups exhibiting maximal treatment-effect heterogeneity while enforcing covariate balance, thus directly controlling confounding without explicitly modeling treatment or outcome mechanisms. The framework accommodates flexible subgroup definitions, allowing additional constraints such as fairness criteria to be incorporated. We show that our approach admits flexible nonparametric estimators and enjoys finite-sample error guarantees. We also introduce a principled rule for subgroup assignment based on observed covariates. Simulated and real-world experiments demonstrate substantial improvements over existing approaches.

## 1 Introduction

### 1.1 Treatment Effect Heterogeneity and Subgroup Analyses

In causal inference, population-level measures, such as the average treatment effect (ATE), have consistently been among the most sought-after effects. However, treatment effects often vary across subpopulations, with some benefiting more, or even being harmed, while others see minimal change. Such heterogeneity can be obscured by population averages (Cintron et al., 2022). Identifying subgroups with distinct treatment responses is essential for advancing scientific insight and informing policy decisions, especially in personalized medicine and healthcare research (Lipkovich et al., 2017).

We study treatment effect heterogeneity from a perspective that differs from conventional approaches. To our knowledge, subgroup analysis in causal inference primarily follows two approaches. The first, based on causal graphs or structural causal models, explores *structural heterogeneity* and the associated subgroup structures (e.g., Kummerfeld & Ramsey, 2016; Hu et al., 2018; Huang et al., 2019; Nagpal et al., 2020; Markham et al., 2022). The second approach emphasizes *treatment effect heterogeneity* within the potential outcome or counterfactual framework, which is widely utilized in statistics and epidemiology. This paper adopts the latter, focusing on effect-based heterogeneity.

To better understand treatment effect heterogeneity, investigators often estimate the conditional average treatment effect (CATE), a widely used estimand that enables personalized treatment assignments based on individual covariate information. Numerous methods have been developed to improve the accuracy and validity of CATE estimation, with recent advances focusing on utilizing supervised machine learning techniques (e.g., Foster et al., 2011; Imai et al., 2013; van der Laan & Luedtke, 2014; Athey & Imbens, 2016; Grimmer et al., 2017; Shalit et al., 2017; Zhang et al., 2017; Künzel et al., 2017; Nie & Wager, 2017; Wager & Athey, 2018; Kennedy, 2020; Zhou & Zhu, 2021). Leveraging accurate CATE estimates, most data-driven subgroup analysis studies identify subgroups where the CATE exceeds a clinically relevant threshold, or through recursive partitions to form a tree whose terminal nodes define the subgroups (e.g., Su et al., 2009; Zhao et al., 2013; Loh et al., 2015; Ondra et al., 2016; Schnell et al., 2016; Chen et al., 2017; Ballarini et al., 2018; Loh et al., 2019; Dwivedi et al., 2020; Hejazi et al., 2021; Qi et al., 2021; Wang & Rudin, 2022).

The CATE is often not the main focus. Instead, the emphasis may be on a lower-dimensional subset of the covariate space $\mathcal{X}$, or on identifying an optimal partition (or subgroup structure) under specific criteria. This area of research remains relatively underexplored, with only a few prior contributions. Kallus (2017) proposed a subgroup partition algorithm for determining a subgroup structure that

minimizes the personalization risk. Building on techniques from unsupervised learning, Kim et al. (2024a;b) introduced a novel framework for investigating heterogeneous treatment effects.

It is often the case that covariate balance between the treated and control units is not guaranteed within the identified subgroups. This imbalance could potentially lead to confounding bias when estimating subgroup effects in observational studies. Similarly, it has been noted that strong performance of a CATE estimator does not necessarily ensure accurate estimation of subgroup effects (Dwivedi et al., 2020). This line of research has received relatively little attention. Recently, some progress has been made to reach a comprise between global and subgroup balance (e.g., Dong et al., 2020; Yang et al., 2021; Ben-Michael et al., 2023). However, these methods require pre-specified subgroups and are not directly applicable to data-driven approaches for identifying effect-based subgroups.

## 1.2 Balancing Approach in Causal Inference

Covariate balancing is central to causal inference in observational studies, where randomized assignment is infeasible. Traditional modeling approaches focus on accurate estimation of the propensity score, while modern balancing methods directly optimize sample weights to ensure covariate balance. It has been shown that balancing methods substantially improve finite-sample performance and yield more stable estimators by minimizing weight dispersion subject to covariate balance constraints. (e.g., Hainmueller, 2012; Zubizarreta, 2015; Chattopadhyay et al., 2020; Ben-Michael et al., 2021). Recent advances incorporating kernel methods into balancing frameworks have enhanced their flexibility and scalability, enabling effective application to high-dimensional settings and complex outcome model spaces. (e.g., Wong & Chan, 2018; Hazlett, 2020; Kim et al., 2024c).

Moreover, since balancing approaches are formulated as optimization problems, they can naturally incorporate practical constraints when defining causal subgroups. One important constraint of this kind is subgroup fairness, the specific case we focus on in this work. It seeks subgroup structures that are approximately independent of sensitive attributes such as race, gender, or socioeconomic status, as measured by a variety of fairness metrics (e.g., Hardt et al., 2016; Corbett-Davies et al., 2017; Barocas et al., 2023). Fairness-aware causal subgroup detection is particularly important when the resulting subgroups are used for downstream policy learning (e.g., Nabi et al., 2019; Viviano & Bradic, 2022; Kim & Zubizarreta, 2023; Suk et al., 2024). From a policymaker's perspective, one might also require that subgroups be defined in terms of other specific geographic or demographic variables. However, a key gap in treatment-effect-based subgroup analysis remains: there is no general framework to ensure that algorithmically discovered subgroups respect such user-specified constraints. We shall show that balancing methods offer a promising approach to jointly control confounding bias and accommodate flexible, user-defined constraints in subgroup analysis.

**Contribution.** We propose a novel optimization-based approach for uncovering treatment-responsive subgroups under direct bias control. The method jointly optimizes subgroup indicators and balancing weights to maximize between-subgroup heterogeneity while enforcing covariate balance and, when required, additional constraints such as fairness. The proposed approach is entirely data-driven and does not require a priori knowledge of subgroup selection criteria. We formulate the method as a mixed-integer quadratic programming problem, which can be solved using readily-available solvers. Notably, bias control is imposed directly at the subgroup level, obviating the need for explicit modeling of the treatment or outcome models. In particular, we show that the subgroups can be found by balancing flexible kernel basis functions in a large nonparametric model, providing consistency guarantees without requiring structural assumptions on the nuisance components. We further describe a procedure for identifying subgroup membership based on individual's observed covariates, which enables our approach to play a prescriptive role in policy settings.

# 2 Framework

## 2.1 Motivating Illustration

We provide an illustrative example in which standard methods fail to adjust covariate imbalance in finite samples and are unable to enforce subgroup fairness. We generate a sample of size $n=2000$ using a simple data-generating process with a covariate $X$, a binary treatment $A$, a binary sensitive

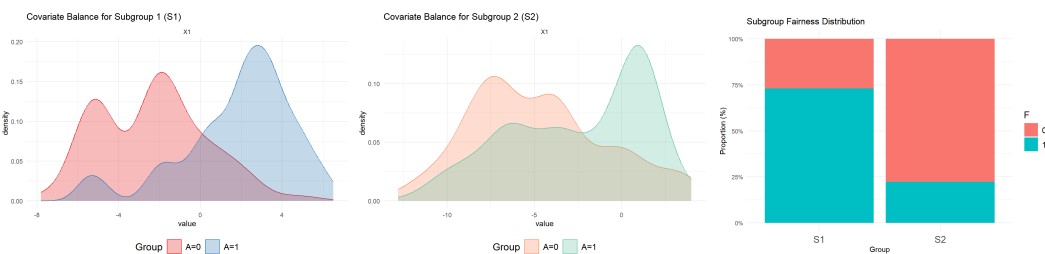

Figure 1: (Left) Positive responders. (Center) Negative responders. (Right) Subgroup unfairness.

variable $F$, and an outcome $Y$, defined as follows:

$$F \sim \text{Bernoulli}(0.5), \; X \sim N(4F - 2, 2), \quad A = \mathbb{1}\{F + 0.5FX + \epsilon_1 > 0\},$$

$$Y = AY^1 + (1 - A)Y^0, \quad Y^1 = X^2 + \Lambda\left(\frac{X - 2}{2}\right) + \epsilon_2, \quad Y^0 = X^2 + \Lambda\left(\frac{X + 2}{2}\right) + \epsilon_2,$$

where we let $Y^a$ denote the potential outcome that would have been observed under the treatment assignment $A = a$, $\Lambda(x) := \max(1 - |x|, 0)$, and $\epsilon_1, \epsilon_2 \sim N(0, 1)$. The population naturally splits into two subgroups characterized by positive and negative treatment effects, respectively.

We aim to identify two subgroups that exhibit similar responses to the treatment $A$ and assess the fairness of each subgroup with respect to the sensitive feature $F$. Following the standard procedure for the effect-based subgroup discovery outlined in Section 1.1, we first estimate the CATE and then identify subgroups of positive and negative responders based on whether their CATE estimates exceed or fall below a threshold $t \geq 0$. The CATE is estimated using the kernel inverse probability weighting estimator (Zhou & Zhu, 2021), and a threshold of $t = 0$ is used. However, we also confirmed that using alternative methods of Athey & Imbens (2016); Wager & Athey (2018); Kennedy (2020) or non-zero thresholds did not significantly alter the results. Covariate balance within each subgroup is assessed using weights derived from the estimated propensity score via logistic regression.

Figure 1 reveal severe covariate imbalance in both subgroups despite the presence of sufficient overlap. This issue is likely to be amplified in real-world settings, where high-dimensional covariates and noise can cause subgroup discovery to capture spurious associations. Moreover, each subgroup exhibits significant fairness concerns, as data-driven policymaking might disproportionately suggest treating patients predominantly characterized by $F = 1$, which may result in policies that reinforce rather than reduce existing disparities. Fairness constraints mitigate this risk by ensuring that "high-benefit" groups are not artifacts of structural inequities and align with policy initiatives aimed at promoting equity. For example, in allocating a costly Medicaid intervention, ignoring historical disparities could result in disproportionate allocation to certain demographic groups even when true effects are comparable across populations. Our approach is designed to prevent such outcomes.

## 2.2 Problem and Setup

Consider a random sample $(Z_1, ..., Z_n)$ of $n$ tuples $Z = (Y, A, F, X)$, where $Y \in \mathbb{R}^p$ represents the outcome, $A \in \{0, 1\}$ denotes a binary intervention, $F \in \{0, 1\}$ is a binary sensitive feature, and $X \in \mathcal{X} \subseteq \mathbb{R}^d$ comprises observed covariates. Here, we accommodate multivariate outcomes; while $p = 1$ is the most common case in practice, we allow $p > 1$. We label the subgroups with the indicator variables $\vec{S} = (S_1, ..., S_R)$, where $S_r = 1$ if a subject belongs to the $r$-th subgroup and $S_r = 0$ otherwise, $r = 1, \ldots, R$. Let $S_{ir}$ denote the subgroup membership for each subject $i = 1, \ldots, n$. In our setting, the number of subgroups $R$ is prespecified, yet each tuple of the subgroup variables $\vec{S}_i \equiv (S_{i1}, \ldots, S_{iR})$ is unknown. For the sake of simplicity, we consider mutually exclusive subgroups, thus $\sum_{r=1}^{R} S_{ir} = 1$. However, this requirement can be relaxed with a minor modification of the optimization problem that we will formulate subsequently.

Throughout, we rely on the following identification assumptions (e.g., Imbens & Rubin, 2015, Chapter 12): (C1) *Consistency*, $Y = Y^a$ if $A = a$; (C2) *No unmeasured confounding*, $A \perp\!\!\!\perp Y^a \mid X, \vec{S}$; (C3) *Positivity*, $\mathbb{P}(A = a | X, \vec{S}) \geq \varepsilon$ a.s. for some $\varepsilon > 0$. Collectively, assumptions (C1) - (C3) imply that

the average treatment effect of the $r$-th subgroup can be identified and expressed in a weighting form as

$$\tau_r = \mathbb{E}(Y^1 - Y^0 \mid S_r = 1) = \mathbb{E}\left\{ \left( \frac{A}{\pi_r(X)} - \frac{1 - A}{1 - \pi_r(X)} \right) Y \;\middle|\; S_r = 1 \right\}, \tag{1}$$

where $\pi_r(X) = \mathbb{P}(A = 1 \mid X, S_r = 1)$ is the *subgroup propensity score*. When $\pi_r$ is known, covariate balance within the r-th subgroup can always be achieved by weighting the treated and control units by $1/\pi_r(X_i)$ and $1/\left(1 - \pi_r(X_i)\right)$, respectively.

**Subgroup covariate balance.** Assuming knowledge of $S_{ir}$ for each $(i, r)$, the subgroup effect $\tau_r$ in 1 can be estimated using the following normalized weighting estimator,

$$\widehat{\tau}_r = \frac{\sum_{i=1}^n A_i S_{ir} w_i Y_i}{\sum_{i=1}^n A_i S_{ir} w_i} - \frac{\sum_{i=1}^n (1 - A_i) S_{ir} w_i Y_i}{\sum_{i=1}^n (1 - A_i) S_{ir} w_i} \tag{2}$$

where $w_i$ are suitable weights (Yang et al., 2021). 2 is often referred to as the Hájek estimator. Recall that condition $S_r = 1$ implies $S_{r'} = 0$ for all $r' \neq r$ due to mutual exclusivity. Arguably, the most common approach to computing the weights is the inverse probability of treatment weighting, where we first model the subgroup propensity score and then inverting the predicted propensities. However, this requires modeling the nuisance function $\pi_r$, making it susceptible to covariate imbalance arising from model misspecification, small samples, or high-dimensional covariates. In addition, even under near-violations of the positivity assumption (C3), this approach can yield highly variable weights and thus produce unstable estimators (Kang & Schafer, 2007). To address these challenges, balancing methods directly compute covariate-balancing weights. In our case, for example, covariate balance in the $r$-th subgroup can be, at least approximately, achieved by enforcing

$$\left| \sum_{i=1}^n A_i S_{ir} w_i \phi_j(X_i) - \sum_{i=1}^n (1 - A_i) S_{ir} w_i \phi_j(X_i) \right| \leq \delta, \tag{3}$$

for a finite set of basis functions $\{\phi_j\}$ and a tolerance level $\delta \geq 0$. Here, the weights $w = \{w_i\}$ are normalized within subgroups: i.e., $\sum_{i=1}^n A_i S_{ir} w_i = \sum_{i=1}^n (1 - A_i) S_{ir} w_i = 1$. In Section 4, we analyze how this balancing condition controls the bias of the estimator $\widehat{\tau}_r$.

**Subgroup fairness.** While our framework is capable of incorporating various types of constraints in subgroup definition, in this work we focus specifically on subgroup fairness. To evaluate unfairness within each subgroup, we utilize the *fairness function* uf $: \mathcal{Y} \times \mathcal{X} \times \{0, 1\}^2 \to \mathbb{R}$, which accommodates a broad range of (counterfactual) fairness measures (e.g., Mishler & Kennedy, 2022; Kim & Zubizarreta, 2023; Suk et al., 2024). Using the fairness function, our subgroup fairness criterion can be expressed at the population level as:

$$|\mathbb{E}\left\{ \text{uf}(Z) \cdot S_r \right\}| \leq \delta', \; \forall r, \tag{4}$$

where $\delta'_r$ represents the acceptable fairness threshold for the r-th subgroup. For instance, the criterion of *independence* or *statistical parity*, arguably one of the most widely recognized fairness criteria, can be implemented by defining the fairness function as:

$$\text{uf}(Z) = \frac{1 - F}{\mathbb{E}(1 - F)} - \frac{F}{\mathbb{E}(F)}, \tag{5}$$

which leads to $|\mathbb{P}\left(S_r = 1 \mid F = 0\right) - \mathbb{P}\left(S_r = 1 \mid F = 1\right)| \leq \delta', \; \forall r$. This requires our identified subgroups to be marginally (approximately) independent of the given sensitive feature. In our estimator, we will use the empirical version of 4 as our fairness constraint: $\left| \mathbb{P}_n \left\{ \widehat{\text{uf}}(Z) \cdot S_r \right\} \right| \leq \delta'$, where we let $\mathbb{P}_n$ denote the empirical measure over $(Z_1, ..., Z_n)$.

The sensitive variable $F$ within each subgroup could be protected using other fairness measures as well, such as conditional statistical parity, equalized odds, or balance for the positive class, with their corresponding fairness functions (see (Mishler & Kennedy, 2022, Section 3) and (Kim & Zubizarreta, 2023, Section 2.3) for examples). Our framework allows for the simultaneous application of multiple fairness measures. The proposed methods remain valid even without subgroup fairness, or other user-defined, constraints. In some real-world subgroup discovery problems, fairness constraints may not be required at all. These constraints are included only to accommodate practical considerations when fairness is a priority.

**Subgroup discovery.** We aim to find subgroups with markedly different responses to a specified treatment relative to the overall population, while maintaining a high degree of homogeneity within each subgroup. To determine subgroup membership and improve interpretability, we define *separation function* $D_{\text{sep}}$ as the sum of pairwise distances between subgroup effects with respect to some distance metric $d : \mathbb{R}^p \times \mathbb{R}^p \to \mathbb{R}^+$, which thus measures the cumulative separation of distinct subgroup effects. Specifically, given a sample of size $n$ and balancing weights $w$, we determine the memberships $\{\vec{S}_i\}_{i=1}^n$ by maximizing the empirical separation:

$$\widehat{D}_{\text{sep}} := \sum_{r \neq r'} d(\widehat{\tau}_r, \widehat{\tau}_{r'}; \{\vec{S}_i\}_{i=1}^n) \tag{6}$$

The (maximum) number of subgroups $R$ could be chosen in advance based on the experimental design or research goal. Practically the most commonly-used cases would be $R = 2$. This is essentially because identifying the most and least benefiting subgroups should be of highest concern to policy-makers. Setting $R = 2$ gives an immediate solution to this problem. However, one may also choose $R$ in a data-driven way, based on the elbow method in a similar spirit to the cluster analysis.

In our framework, subgroups are identified by maximizing empirical separation while simultaneously ensuring covariate balance and, if necessary, fairness conditions within each subgroup. Ours can be viewed as a more generalized, distribution-free version of targeting subgroup through a scoring system, i.e., thresholding the CATE function (e.g., Zhao et al., 2013; Wang & Rudin, 2022). The proposed approach can be viewed as a generative model for heterogeneous treatment effects, treating subgroup indicators as random variables drawn from a distribution that maximizes separation.

## 3 ESTIMATION

For estimation, we jointly optimize three objectives in a single optimization process: covariate balance (3), subgroup fairness (4), and maximal separation of subgroup effects (6).

**Constraints.** Our decision variables consist of two components; the integer variables $\vec{S}_1, \ldots, \vec{S}_n$, $\vec{S}_r \in \{0, 1\}^R$, $r = 1, \ldots, R$, which determine subgroup membership, and the continuous variables $w = (w_i, \ldots, w_n)$, $w_i \in [0, 1]$, which are the weights of each observation. The constraints associated with our causal subgroup discovery problem are:

$$\widehat{\tau}_r = \sum_i A_i S_{ir} w_i Y_i - \sum_i (1 - A_i) S_{ir} w_i Y_i, \; \forall r, \tag{7a}$$

$$\sum_i A_i S_{ir} w_{ir} = \sum_i (1 - A_i) S_{ir} w_{ir} = 1, \; \forall r, \tag{7b}$$

$$\max_{j \in \{1, \ldots, B\}} \Big| \sum_i A_i S_{ir} w_i \phi_j(X_i) - \sum_i (1 - A_i) S_{ir} w_i \phi_j(X_i) \Big| \leq \delta, \; \forall r, \tag{7c}$$

$$\Big| \mathbb{P}_n \Big\{ \widehat{\text{uf}}(Z) \cdot S_r \Big\} \Big| \leq \delta', \forall r, \tag{7d}$$

$$\sum_i S_{ir} A_i \geq n_{min}, \quad \sum_i S_{ir} (1 - A_i) \geq n_{min}, \; \forall r, \tag{7e}$$

$$\sum_r S_{ir} = 1, \; \forall i, \tag{7f}$$

$$w_i \geq 0, \; \forall i \tag{7g}$$

(7a) and (7b) together define the subgroup effect estimator $\widehat{\tau}_r$, $r = 1, \ldots, R$. As discussed in the previous section, (7c) ensures that covariate balance is achieved within each subgroup. It is desirable to choose a flexible set of basis functions $\{\phi_1, \ldots, \phi_K\}$ that spans a general model space for the response surface; examples include power series, kernel, splines. For example, building on recent work (Hazlett, 2020; Kim et al., 2024c), we may employ a kernel basis by setting $\phi_j(X_i) = K(X_j, X_i)$ for a kernel (Gram) matrix $\boldsymbol{K}$. This enables flexible balancing in a reproducing kernel Hilbert space, with further details provided in the next section.

The subgroup fairness constraints (7d) depend on the choice of fairness measures, and thereby corresponding fairness functions. One of the most widely used group-fairness criteria is statistical parity,

which gives $\mathbb{P}_n \left\{ \widehat{\mathrm{uf}} \cdot S_r \right\} = \mathbb{P}_n \left[ \left\{ \frac{(1-F)}{\mathbb{P}_n(1-F)} - \frac{F}{\mathbb{P}_n(F)} \right\} S_r \right]$. This sample-average type estimator converges quickly at root-n rates to the original population-level functional 4. Note that in our setting, multiple fairness constraints can be employed, such as $\mathrm{uf}_1$, $\mathrm{uf}_2$, and so on.

The tolerance levels are usually determined by user beforehand. When feasible in the data setting, investigators always can choose smaller values for $\delta$, $\delta'$ hoping to reduce imbalances. (7e) ensures that the number of treated and control units within each subgroup is at least $n_{min}$, which typically increases with $n$. This could be useful when we want to avoid formation of extremely small-size subgroups. (7f) is used for mutual exclusivity across subgroups. Finally, (7g) restricts the weights to be positive, which forces all weight-based adjustments to be an interpolation as opposed to an extrapolation of the observed data (Zubizarreta, 2015).

**Objectives.** The objective function to be maximized consists of the empirical separation function 6 and a weight regularization term $\Omega(w)$ with penalty parameter $\lambda \geq 0$:

$$\sum_{r \neq r'} d(\widehat{\tau}_r, \widehat{\tau}_{r'}; \{\vec{S}_i\}_{i=1}^n) - \lambda\Omega(w).$$

A natural choice for $d$ is $L_q$-norm, $q \geq 1$, which reduces to the absolute value function for univariate outcomes. The regularization term is set to restrain the variability in the weights, thereby preventing extreme weights (Chattopadhyay et al., 2020). Different choices of $\Omega(\cdot)$ have been used in the balancing literature; for instance, Hainmueller (2012) used the Kullback entropy divergence and Zubizarreta (2015) used the sum of the squared weights. We set $\Omega(w) = \sum_{i=1}^n w_i^2$ based on the theoretical analysis presented in Section 4.

Our estimator can be formulated as a mixed-integer quadratic program. Increasing $R$ may lead to a considerable rise in computational cost. So we propose an alternative sequential procedure for practical implementation in a spirit similar to tree-based modeling: we begin by solving the problem with $R = 2$ (the most interpretable case) and iteratively identify additional splits within each subgroup until a predefined stopping criterion is met (e.g., infeasibility). More effective heuristics should be explored in future work.

## 4 ANALYSIS

### 4.1 ADDITIVE MODEL

Given the pivotal role of the subgroup estimator $\widehat{\tau}_r$ in (7a) within our proposed subgroup discovery procedure, it is essential to theoretically assess its estimation accuracy for any given subgroup membership indicators. Unlike conventional modeling approaches, our method does not rely on any modeling assumptions for nuisance functions such as $\pi_r$. In the balancing approach, the performance of estimator rather critically depends on the function space for the outcome surface, spanned by the basis set $\{\phi_1, \ldots, \phi_B\}$. As in Yang et al. (2021), one may start with positing the following simple parametric additive models where the treatment effect is homogeneous within a subgroup:

$$Y^a = \sum_{r=1}^R \beta_r S_r + \sum_{r=1}^R S_r \sum_{j=1}^B \beta_{rj} \phi_j(X) + a \sum_{r=1}^R \tau_r S_r + \epsilon_a, \quad \forall a \in \{0, 1\}, \tag{8}$$

where $\epsilon_a$ is a mean-zero random vector in $\mathbb{R}^p$. If the normalization and balancing conditions (7c), (7b) hold, then by (Yang et al., 2021, Proposition 2), one may bound the bias for $\widehat{\tau}_r$ in terms of $\delta$, i.e., $\|\mathbb{E}(\widehat{\tau}_r - \tau_r)\|_1 \leq \delta \sum_{j=1}^B \|\beta_{rj}\|_1$, where $\|\cdot\|_q$ denotes the $L_q$-norm. In the next theorem, we go one step further and derive error bounds for the subgroup weighting estimator (2).

**Theorem 4.1.** *Suppose that the outcome surface satisfies the additive model 8, where the variance of each noise element $\epsilon_a$ is finitely bounded by $\sigma_\infty^2$. Given the subgroup membership $\vec{S}_1, \ldots, \vec{S}_n$ and any weights $w$ that satisfy the conditions delineated in (7a), (7b), and (7c), when the pairwise distance $d$ and the separation $D_{sep} = \sum_{r \neq r'} d(\tau_r, \tau_{r'})$ are defined with respect to the $L_q$ distance, it follows that*

$$\mathbb{E}\{d(\tau_r, \widehat{\tau}_r)\} \leq \delta \sum_{j=1}^B \|\beta_{rj}\|_q + p\sigma_\infty \sqrt{n \sum_{i=1}^n w_i^2},$$

$$\mathbb{E}\left|\widehat{D}_{sep} - D_{sep}\right| \leq \delta R \left(\sum_{r=1}^{R}\sum_{j=1}^{B}\|\beta_{rj}\|_q\right) + p\sigma_\infty R^2 \sqrt{n\sum_{i=1}^{n} w_i^2}.$$

*If we further assume that $\{\epsilon_{ai}\}_{i=1}^{n}$ are independent sub-Gaussian random vectors with parameter $\sigma_a$, then with probability at least $1 - \xi$, $\xi > 0$, we have that*

$$d(\tau_r, \widehat{\tau}_r) \leq \delta \sum_{j=1}^{B}\|\beta_{rj}\|_q + p\left(\sigma_0 + \sigma_1\right)\sqrt{2\log\left(\frac{2}{\xi}\right)\sum_{i=1}^{n}w_i^2},$$

$$\left|\widehat{D}_{sep} - D_{sep}\right| \leq \delta R \sum_{r=1}^{R}\sum_{j=1}^{B}\|\beta_{rj}\|_q + R^2 p\left(\sigma_0 + \sigma_1\right)\sqrt{2\log\left(\frac{2}{\xi}\right)\sum_{i=1}^{n}w_i^2}.$$

## 4.2 KERNEL BASIS

The outcome surface could be considered within a substantially larger nonparametric model, avoiding the need to specify structural components as in the additive model in 8. In this subsection, we explore such possibilities based on kernel methods. The additive model 8 has certain drawbacks: (1) it requires explicitly specifying the functional relationship between the potential outcome and the basis functions, and (2) it assumes a constant treatment effect within each subgroup. Here, we attempt to address the first limitation by employing the kernel basis. By Moore-Aronszajn, for every symmetric, positive definite kernel $K : \mathcal{X} \times \mathcal{X} \to \mathbb{R}$, there exists a unique reproducing kernel Hilbert space (RKHS) $\mathcal{H}_K$ associated with the kernel $K$. Given our choice of such a kernel $K$, we construct the kernel matrix $\boldsymbol{K}$ of size $n \times n$, then let $\phi_j(X_i) = \boldsymbol{K}_{ij}$. This enables to consider a more flexible outcome surface model, as formally stated in the following theorem.

**Theorem 4.2.** *Suppose that the outcome surface satisfies*

$$Y^a = \sum_{r=1}^{R} S_r m(X, S_r = 1) + a\sum_{r=1}^{R}\tau_r S_r + \epsilon_a, \ \forall a \in \{0, 1\}, \tag{9}$$

*for each $m(\cdot, S_r = 1) \in \mathcal{H}_K$, an RKHS induced by a Mercer kernel $K$, where $var(\epsilon_a) < \sigma_\infty^2$. We assume that $\{\epsilon_{ai}\}_{i=1}^{n}$ are independent sub-Gaussian random vectors with parameter $\sigma_a$, and that we draw $X_1, \ldots, X_n$ i.i.d. from $\mathbb{P}$ where $\mathbb{P}$ has full support on $\mathcal{X}$. Then given the subgroup membership $\vec{S}_1, \ldots, \vec{S}_n$ and any weights $w$ that satisfy the conditions in 7a, 7b, 7c, we obtain that, for some $\{\alpha_i\}$ depending on $n$ and $\{X_i\}$,*

$$d(\tau_r, \widehat{\tau}_r) = O\left(\delta\sqrt{\sum_{j=1}^{n}\alpha_j^2}\right) + O_{\mathbb{P}}\left(\sqrt{\sum_{i=1}^{n}w_i^2}\right) + o_{\mathbb{P}}(1),$$

*and $d(\tau_r, \widehat{\tau}_r) \asymp \left|\widehat{D}_{sep} - D_{sep}\right|$,*

Next, we tackle the second limitation by fully relaxing the assumption of a constant treatment effect within each subgroup, allowing for greater flexibility in modeling treatment heterogeneity. Specifically, we posit the outcome surface

$$Y^a = \mathbb{E}(Y^a \mid X, \vec{S}) + \epsilon_a = \mu_{A=a, \vec{S}}(X) + \epsilon_a \tag{10}$$

Then we have the identifying expression for $\tau_r$ as

$$\tau_r = \mathbb{E}\left\{\mu_{1,r}(X) - \mu_{0,r}(X) \mid S_r = 1\right\},$$

where $\mu_{a,r}(X) \equiv \mu_{A=a, S_r=1}(X)$. In what follows, we show that when $\mu_{a,r} \in \mathcal{H}_K$, it is possible to achieve error bounds comparable to those established in Theorem 4.2.

**Theorem 4.3.** *Suppose that the outcome surface satisfies 10 where $\mu_{a,r} \in \mathcal{H}_K$ and $var(\epsilon_a) < \sigma_\infty^2$. We adopt the same assumptions as in Theorem 4.2, except that, in place of 7c, we impose the following balancing condition:*

$$\max_j \left|\sum_i \mathbb{1}(A_i = a)S_{ir}w_i\boldsymbol{K}_{ij} - \frac{\frac{1}{n}\sum_{i=1}^{n}S_{ir}\boldsymbol{K}_{ij}}{\mathbb{P}_n(S_r)}\right| \leq \delta, \tag{11}$$

$\forall a, r$. *Then, for some* $\{\alpha_i\}$ *depending on* $n$ *and* $\{X_i\}$, *we have* $d(\tau_r, \widehat{\tau}_r) = O\left(\delta\sqrt{\sum_{j=1}^n \alpha_j^2}\right) +$ $O_{\mathbb{P}}\left(\sqrt{\sum_{i=1}^n w_i^2}\right) + o_{\mathbb{P}}(1)$ *and* $d(\tau_r, \widehat{\tau}_r) \asymp \left|\widehat{D}_{sep} - D_{sep}\right|$,

Condition 11 ensures that the weighted sample kernel mean of treated/control units within the subgroup closely approximates the kernel mean of the subgroup's target population. Both 9 and 10 have not been previously considered in the context of subgroup analysis. Especially, 10 enables significantly more flexible modeling of the outcome surface. However, incorporating the balancing condition 11 into the constraint set introduces significant non-linearity to the optimization problem, potentially leading to increased computational complexity. Thus, we propose solving a relaxed problem by fixing $\mathbb{P}_n(S_r) = z_r$, $m \leq z_r \leq n - (R-1)m$ for $r = 1, \ldots, R$, evaluating the optimization at discrete points on the grid $\{z_r\}_{r=1}^R \subseteq [m, n - (R-1)m]^R$.

Theorems 4.1 - 4.3 establish error bounds for $\widehat{\tau}_r$ and $\widehat{D}_{\text{sep}}$, considering different outcome surface models. Importantly, the finite-sample error bounds we derive hold uniformly over all subgroup–weight pairs that satisfy the constraints. Consequently, inference is anchored to the entire feasible set rather than to a single, ex-post selected subgroup. In other words, we do not require the subgroup indicators to be fixed in advance; instead, we control estimation error uniformly over all admissible $S$, thereby mitigating the risk of selective inference and overfitting that can arise in classical constrained M-estimation. They also motivate our choice of weight regularization as $\Omega(w) = \sum_{i=1}^n w_i^2$.

## 4.3 ESTIMATING SUBGROUP MEMBERSHIP

In subgroup analysis, it is also essential to determine subgroup membership for a new individual based on their observed covariates. In this subsection, we outline a simple nonparametric approach for identifying subgroup membership by estimating the *membership probability* $\mathbb{P}(S_r = 1 \mid X)$. We first construct the kernel density estimator of $p_r(x) \equiv \mathbb{P}(X = x \mid S_r = 1)$ given by the formula

$$\widehat{p}_{r,h}(x) = \frac{\sum_{i=1}^n T_{h,x}(X_i)\mathbb{1}(S_{ir} = 1)}{n_r}\mathbb{1}(n_r > 0), \tag{12}$$

where we let $T_{h,x} = \frac{1}{h^d}k\left(\frac{\|X - x\|_2}{h}\right)$ and $n_r = \sum_{i=1}^n \mathbb{1}(S_{ir} = 1)$, with a kernel function $k : \mathbb{R}^d \to \mathbb{R}$ that is an integrable function satisfying $\int K(u)du = 1$ and the bandwidth $h$. Then the the Bayes' theorem suggests the following estimator for $\mathbb{P}(S_r = 1 \mid X)$:

$$\widehat{\mathbb{P}}(S_r = 1 \mid X = x) = \frac{\widehat{p}_{r,h}(x)\mathbb{P}_n(S_r)}{\sum_{r'} \widehat{p}_{r',h}(x)\mathbb{P}_n(S_{r'})}. \tag{13}$$

Note that $\mathbb{P}_n(S_r)$, $n_r$ are known. Once we have computed 13 for $r = 1, \ldots, R$, we may determine the subgroup membership for an individual with $X = x$ as

$$\widehat{r}^* = \underset{r \in \{1, \ldots, R\}}{\arg\max} \widehat{\mathbb{P}}(S_r = 1 \mid X = x),$$

which allows our proposed method to play a prescriptive role in policy making as well. The estimated membership probability could be used as a confidence level when making recommendations on a new sample. The next proposition gives conditions under which the proposed estimator to determine subgroup membership is consistent.

**Proposition 4.4.** *Consider a probability distribution* $\mathbb{P}$ *from which a set of independent random vectors* $\{(X_i, A_i, Y_i, \vec{S}_i)\}_{i=1}^n$ *is drawn, each having a common mean and finite second and third central moments. Assume that the conditional distribution of* $X$ *given* $S_r = 1$ *is absolutely continuous with respect to the Lebesgue measure on* $\mathbb{R}^d$. *Let* $p_r$ *be a density function in the Hölder class with parameters* $(\beta, L)$, *and* $k$ *be a kernel of order* $\lfloor \beta \rfloor$ *satisfying* $\int \|u\|_1^\beta |k(u)|du < \infty$ *and* $\|k\|_\infty < \infty$. *If* $n^{-1}h^{-d} + h^{2\beta} = o(1)$ *and* $\mathbb{P}_n\{S_r\} \xrightarrow{p} \mathbb{P}(S_r = 1)$, *then the proposed estimator 13 is consistent, and*

$$\mathbb{P}(\widehat{r}^* \neq r^*) \to 0,$$

*where* $r^* = \underset{r}{\arg\max} \mathbb{P}(S_r = 1 \mid X = x)$.

See Definition B.1 in Appendix B for the formal definition of the Hölder class density functions and high-order kernels.

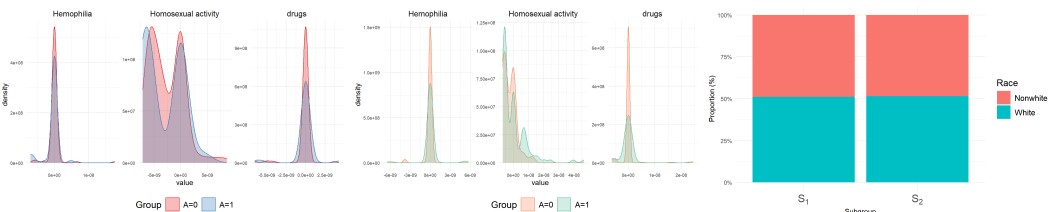

Figure 2: Distributions of risk fac-Figure 3: Distributions of risk fac-Figure 4: Ethnic group distribu-
tors for $S_1$ after adjustment.     tors for $S_2$ after adjustment.     tions

| Subgroup (effect) | Hemophilia | Homosexuality | Drug use | White/Non-White ratio |
|---|---|---|---|---|
| $S_1$ (+118) | 0.10 (0.09) | 0.67 (0.22) | 0.13 (0.10) | 1.05 |
| $S_2$ (−85) | 0.05 (0.05) | 0.42 (0.25) | 0.06 (0.04) | 1.07 |

Table 1: Estimated subgroup effects and key summary characteristics across subgroups.

## 5 EXPERIMENTS

### 5.1 SIMULATION STUDY

By extending our motivating example in Section 2.1, we conduct a simulation study to illustrate
the proposed methods, where we compare the proposed methods with widely-used CATE-based
approaches under both correctly specified and misspecified covariate settings, while also assessing
their ability to satisfy the required fairness criterion. The results demonstrate that the proposed
methods not only achieve the desired level of fairness within subgroups but also provide more
accurate subgroup effect estimates compared to the baselines. Due to the page limit, we defer the
detailed simulation setup and results to Appendix A.1.

### 5.2 CASE STUDY

Here, we apply our proposed methods to analyze the effect of combined antiretroviral therapy in
the treatment of HIV. The data used in this analysis are derived from the ACTG 175 randomized
trial (Hammer et al., 1996). The treatment is whether patients received combination therapy ($A = 1$)
versus zidovudine alone ($A = 0$), and the outcome is CD4 count. Our objective is to identify two
subgroups in which the combination therapy shows the greatest ($S_1$) and least ($S_2$) effectiveness,
i.e., an increase in the CD4 count. Just for illustrative purposes, we assume that ethnicity has no
genetic effect on treatment efficacy and seek to ensure the absence of ethnicity-related bias within
each subgroup. Figures 2 and 3 show the distribution of the three key risk factors after adjustment,
indicating that covariates are well balanced within each subgroup under the estimated weights, thereby
ensuring negligible bias. Figure 4 presents the ethnicity distributions across subgroups, confirming
that the fairness criterion is met. We present the estimated subgroup effects, along with the mean and
variance of key risk factors in Table 1. We observe substantial effect heterogeneity, with estimated
subgroup effects of 118 for $S_1$ and -85 for $S_2$, aligning with the findings of Kennedy et al. (2023).
Table 1 suggests that the inferior performance of subgroup $S_2$ may be partly attributable to its lower
prevalence of hemophilia, homosexuality, and drug use. More results can be found in Appendix A.2.

## 6 LIMITATIONS

Two caveats merit discussion. First, our results do not fully resolve issues of valid post-selection
inference. Although our finite-sample bounds hold uniformly over all admissible subgroup–weight
pairs, they do not constitute selective $p$-values. This limitation is shared with the broader balancing
literature and remains an open challenge in our framework. This is the price we pay to gain the key
advantage of our framework: bias and fairness can be directly and flexibly controlled without explicit
CATE modeling. Second, there is a trade-off between enforcing balance/fairness and weight stability,
governed by tuning parameters $(\delta, \delta', \lambda)$, whose optimal choice remains open, particularly when
additional constraints are included. Future work should provide theoretical guidance on these issues.

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
