# OpenReview forum: "Uncovering Treatment-Responsive Subgroups Under Direct Bias Control"
_ICLR.cc/2026/Conference — Submitted to ICLR 2026_

### Official Review · Reviewer_9QMm · 2025-10-31

**Soundness:** 3
**Presentation:** 3
**Contribution:** 3
**Rating:** 4
**Confidence:** 4

**Summary:**

This paper proposes an optimization-based framework for discovering treatment-responsive subgroups from observational data while directly controlling confounding bias and fairness constraints. Unlike traditional approaches that rely on estimating CATE or propensity scores, the method jointly optimizes subgroup indicators and balancing weights to achieve covariate balance and fairness while maximizing subgroup heterogeneity.
The authors formalize this as a mixed-integer quadratic programming problem and derive finite-sample error bounds under both additive and kernel-based outcome models. They also propose a nonparametric estimator for subgroup membership probabilities for new samples. Simulation studies and an application to the ACTG 175 HIV trial demonstrate the method's ability to balance covariates and enforce subgroup fairness.

**Strengths:**

1. Timely topic: Addresses an important and underexplored intersection between causal inference, subgroup analysis, and fairness.
2. Solid theoretical development: Provides finite-sample error bounds and a rigorous treatment under both additive and RKHS models.
3. Flexible and interpretable formulation: The optimization-based design allows incorporating userdefined constraints, offering potential policy relevance.
4. Fairness consideration: Explicitly integrates fairness constraints into subgroup discovery, an aspect often neglected in causal subgroup analysis.
5. Empirical demonstration: The ACTG 175 case study nicely illustrates potential policy applications.

**Weaknesses:**

1. Marginal methodological novelty: The proposed framework largely extends known balancing approaches to subgroup discovery, with limited conceptual innovation.
2. Scalability concerns: The mixed-integer quadratic formulation is computationally intensive and may not scale to modern large-scale datasets.
3. Insufficient empirical validation: The experiments are narrow in scope-only one real dataset and limited baselines are tested. No comparison to modern meta-learners or causal clustering methods (e.g., Causal K-means, R-learner variants).
4. Unclear practical benefit: It remains uncertain whether direct bias control meaningfully improves subgroup interpretability or fairness in complex, high-dimensional settings.
5. Theoretical assumptions: Some conditions (e.g., subgroup mutual exclusivity, uniform balance tolerance $\delta$ ) are unrealistic in practical observational studies.

**Questions:**

Please refer to the Weaknesses above.

---

### Official Review · Reviewer_vuC6 · 2025-10-31

**Soundness:** 3
**Presentation:** 3
**Contribution:** 3
**Rating:** 4
**Confidence:** 3

**Summary:**

This paper proposes a new optimization-based framework for discovering treatment-responsive subgroups that maximizes effect heterogeneity while directly enforcing covariate balance and fairness constraints. The work is timely and relevant to causal inference, fairness, and machine learning communities. The contribution lies at the intersection of subgroup discovery and balancing-based causal inference, bridging an important gap between CATE-driven and optimization-based methods. The paper is technically strong, well-positioned with related literature, and includes both finite-sample guarantees and policy-level interpretability through subgroup assignment rules.

**Strengths:**

1. I pretty much enjoy the idea of introducing the idea of optimization into causal inference literature. The framework unifies subgroup discovery, balancing, and fairness via a single optimization problem. The idea is generally elegant and interesting to me.
2. The theoretical results are solid and insightful.
3. The paper is well structured and very easy to follow.

**Weaknesses:**

1. The introduction motivates the need for subgroup discovery but does not clearly distinguish this work from recent causal clustering methods. In the case study, is it possible to show some numerical results showing the advantage of the proposed method? Or, maybe the authors can consider adding a concise table or figure contrasting the different methods, highlighting differences in modeling assumptions, bias control, and fairness.
2. A central methodological concern is that subgroup indicators and weights are optimized jointly from the data, which violates the iid assumptions required for standard asymptotic inference. This limitation is acknowledged by the authors, but it substantially constrains the interpretability of the results in policy or scientific settings.
3. Minor comments: what is $X^2$ in line 123-124?

**Questions:**

See above.

---

### Official Review · Reviewer_DKLZ · 2025-11-01

**Soundness:** 3
**Presentation:** 4
**Contribution:** 3
**Rating:** 4
**Confidence:** 2

**Summary:**

The paper introduces an optimization-based pipeline for discovering various treatment-responsive population subgroups under multiple constraints. The method jointly maximizes between-subgroup heterogeneity while maintaining covariate balance, and optionally enforcing one or more additional constraints such as fairness measures (e.g. statistical parity). The optimization problem is formulated as a solvable MIQP. The method avoids subgroup-specific computation of nuisance functions which could fail in small sample regimes. Consistency guarantees are provided for a nonparametric additive outcome model. The objective is illustrated with simulations and a real data case study.

**Strengths:**

- The paper is very clearly written and well-organized. Theoretical results are sound, and the necessary assumptions are clearly formulated.
- The subgroups of interest can be discovered by optimizing a single optimization objective which directly enforces covariate balance and other optional constraints.
- The framework is modular and flexible: balance can be enforced over user-chosen basis functions (including kernels) and augmented with  fairness or size constraints, making it adaptable to different domains and objectives.

**Weaknesses:**

- The number of groups needs to be specified by the user. For R > 2 a discussion of how misspecification of R affects the method and the optimization seems to be lacking. Additionally, the resulting groups might not be interpretable for a practitioner, however, it is unclear how interpretability can be enforced by the method.
- Considering the number of constraints, it is unclear how often the problem is feasible. The authors could provide a discussion and an additional empirical analysis of settings where no parameters  satisfy all constraints, and discuss what can be done in this setting. This seems to be especially important if more and more constraints have to be satisfied at the same time.
- It would be useful to precisely analyze the runtime / computational complexity of the MIQP in $R$, the number of (fairness) constraints, the number of basis functions, the number of observations and further parameters of the problem. Right now it is unclear how computationally feasible the objective really is, especially in real-data settings.
- Although "data-driven", the method still requires choosing parameters such as $\delta, \delta', n_{min}, R$ and $\gamma$. It would be useful to provide a full algorithm together with instructions how to pick these parameters. For instance, what would the method discover if one of the "true" treatment response groups is underrepresented in the data ($< n_{min}$)? How robust is the procedure against misspecification of these parameters?
- Analysis of overlapping subgroups appears to be lacking.
- In total, although the problem appears interesting, given the narrow scope of the question, I feel like for full soundness the paper would benefit from 1) a more thorough motivation of the problem and more real-world examples; 2) structured ablations and analysis of robustness against misspecification; 3) runtime analysis; 4) more real-world experiments.

**Questions:**

- Could you elaborate on the computational complexity of the method as mentioned above?
- Could you provide an analysis of feasibility as the number of constraints increases?
- Is there a systematic way to choose the hyperparameters $\delta,\delta', n_{min}, R$ and $\gamma$?
- Can the framework be extended to overlapping groups?
- Can the groups be made interpretable?

---

### Meta-Review · Area_Chair_67vL · 2026-01-07

**Summary:**

This paper presents an optimization-based framework for identifying treatment-responsive subgroups by jointly optimizing subgroup indicators and balancing weights. The method maximizes heterogeneity while enforcing covariate balance and optional fairness constraints through a Mixed-Integer Quadratic Programming (MIQP) formulation. The authors provide finite-sample error guarantees and validate the approach using simulated data and a clinical trial case study.

**Reviewer Concerns:**

The primary concerns involve the computational scalability of the MIQP solver for large datasets, the limited empirical comparison against modern causal clustering and meta-learning baselines, and a lack of guidance regarding hyperparameter selection

**Reviewer Scores:**

All three reviewers assigned a score of 4, consistently noting that while the framework is elegant and theoretically solid, its practical limitations and narrow experimental scope currently place it below the acceptance threshold.

---

### Decision · Program_Chairs · 2026-01-26

Reject